# *SLCO1B1* Exome Sequencing and Statin Treatment Response in 64,000 UK Biobank Patients

**DOI:** 10.3390/ijms25084426

**Published:** 2024-04-17

**Authors:** Deniz Türkmen, Jack Bowden, Jane A. H. Masoli, David Melzer, Luke C. Pilling

**Affiliations:** 1Epidemiology & Public Health Group, Department of Clinical & Biomedical Science, Faculty of Health & Life Sciences, University of Exeter, Exeter EX4 4QD, UK; dt381@exeter.ac.uk (D.T.); j.masoli@exeter.ac.uk (J.A.H.M.); d.melzer@exeter.ac.uk (D.M.); 2Exeter Diabetes Group (ExCEED), Department of Clinical & Biomedical Science, Faculty of Health & Life Sciences, University of Exeter, Exeter EX4 4QD, UK; j.bowden2@exeter.ac.uk; 3Department of Genetics, Novo Nordisk Research Centre Oxford, Innovation Building, Old Road Campus, Oxford OX3 7BN, UK; 4Department of Healthcare for Older People, Royal Devon University Healthcare NHS Foundation Trust, Exeter EX2 5DW, UK

**Keywords:** cholesterol, statin, *SLCO1B1*, clinical response, pharmacogenomics, exome sequencing variants, epidemiology

## Abstract

The solute carrier organic anion transporter family member 1B1 (*SLCO1B1*) encodes the organic anion-transporting polypeptide 1B1 (OATP1B1 protein) that transports statins to liver cells. Common genetic variants in *SLCO1B1*, such as *5, cause altered systemic exposure to statins and therefore affect statin outcomes, with potential pharmacogenetic applications; yet, evidence is inconclusive. We studied common and rare *SLCO1B1* variants in up to 64,000 patients from UK Biobank prescribed simvastatin or atorvastatin, combining whole-exome sequencing data with up to 25-year routine clinical records. We studied 51 predicted gain/loss-of-function variants affecting OATP1B1. Both *SLCO1B1**5 alone and the *SLCO1B1**15 haplotype increased LDL during treatment (beta*5 = 0.08 mmol/L, *p* = 6 × 10^−8^; beta*15 = 0.03 mmol/L, *p* = 3 × 10^−4^), as did the likelihood of discontinuing statin prescriptions (hazard ratio*5 = 1.12, *p* = 0.04; HR*15 = 1.05, *p* = 0.04). *SLCO1B1**15 and *SLCO1B1**20 increased the risk of General Practice (GP)-diagnosed muscle symptoms (HR*15 = 1.22, *p* = 0.003; HR*20 = 1.25, *p* = 0.01). We estimated that genotype-guided prescribing could potentially prevent 18% and 10% of GP-diagnosed muscle symptoms experienced by statin patients, with *15 and *20, respectively. The remaining common variants were not individually significant. Rare variants in *SLCO1B1* increased LDL in statin users by up to 1.05 mmol/L, but replication is needed. We conclude that genotype-guided treatment could reduce GP-diagnosed muscle symptoms in statin patients; incorporating further *SLCO1B1* variants into clinical prediction scores could improve LDL control and decrease adverse events, including discontinuation.

## 1. Introduction

*SLCO1B1* (solute organic anion transporter family member 1B) encodes the ‘OATP1B1’ protein (the organic anion-transporting polypeptide 1B1) transporter to liver cells for bilirubin and statins [1]. *SLCO1B1* is therefore classified as ‘very important’ on the pharmacogenetics review site PharmGKB [2]. The common variants *SLCO1B1**5 (rs4149056:Val174Ala) and *SLCO1B1**1B (rs2306283:Asn130Asp—now designated *SLCO1B1**37) have European allele frequencies of ~2% and 40%, with frequencies in other populations varying from <1% to 75% [3]. These variants affect statin pharmacokinetics [1], with *5 linked to increased adverse events [4]. However, some studies report increased statin-associated muscle symptoms (SAMS) for *SLCO1B1**5 [5,6,7,8,9], but others do not [10,11,12] and have found inconclusive trends [13]. The *SLCO1B1* genotype is reported to increase SAMS risk in patients prescribed simvastatin or atorvastatin, but has a smaller effect on pravastatin, rosuvastatin, and fluvastatin [14,15,16]. There are also conflicting reports on haplotype *15 (allele frequency 3–24%) [3] and increasing SAMS [17].

Common *SLCO1B1* “gain-of-function” (GoF) variants (rs11045819:Pro155Thr and rs34671512:Leu643Phe) increase simvastatin bioavailability [18], but their effects on the statin response remain unclear. A recent study in 15,378 statin users combined four common *SLCO1B1* variants into a gene risk score; high-risk patients had increased odds of statin intolerance, and accounting for GoF variants improved prediction [19].

Studies [20,21,22] also identified rare *SLCO1B1* variants. A study of 699 patients using exome sequencing found that rare variants explained 17.8% of *SLCO1B1* variability [23], and ~9% of *SLCO1B1* variants are deleterious rare variants [20,22]. However, the impact on the statin treatment response is unknown due to the widely used genotyping technologies (microarray, imputed) and small sample sizes with exome data.

We aimed to estimate the associations between common and rare *SLCO1B1* variants with statin-related outcomes using data from 64,000 atorvastatin and simvastatin patients in UK Biobank with whole-exome sequencing (WES) data linked to primary and secondary care records. The outcomes were low-density lipoprotein (LDL) levels, General Practice-diagnosed muscle symptoms and treatment discontinuation, and hospital-diagnosed coronary heart disease and stroke. Other biomarkers investigated related to *SLCO1B1* from the GWAS (Genome-Wide Association Study) Catalog included vitamin D (risk factor for muscle weakness [24]) and bilirubin (possible risk factor for increased stroke [25,26]).

## 2. Results

A total of 67,630 participants self-reported simvastatin or atorvastatin use at the baseline assessment (Figure 1); the mean age was 61.5 years, and 38.7% were female (Table 1). A total of 65,513 participants were prescribed simvastatin or atorvastatin by their GP; the mean age at first prescription was 61.0 years, and 41.7% were female. See the Appendix A for details on the overlap.

### 2.1. Common SLCO1B1 Variants

#### Baseline Self-Reported Statin Use and Biomarkers

**LDL cholesterol**: The mean LDL-c level was 2.75 mmol/L (SD 0.66) (2.86 mmol/L (SD 0.67) in women and 2.62 mmol/L (SD 0.65) in men. In the multivariable analysis, two variants were associated with higher LDL levels versus non-carriers: *SLCO1B1**5 (Coef: 0.08, 95%CI: 0.05 to 0.12, *p* = 6 × 10^−6^) and *SLCO1B1**15 (coefficient: 0.03, 95%CI: 0.01 to 0.04, *p* = 3 × 10^−4^) (Figure 2, Table 2). *SLCO1B1**14 carriers (including rs11045819 and rs2306283) had a trend of a reduction in LDL levels (Coef: −0.01, *p* = 0.05).

**Other biomarkers**: While *SLCO1B1**5 and *15 had decreased vitamin D levels (Coef: −2.73, 95%CI: −3.89 to −1.59, *p* = 3 × 10^−6^ and Coef: −0.56, 95%CI: −1 to −0.1, *p* = 0.02, respectively), *SLCO1B1**14 and *20 had increased vitamin D levels vs. non-carriers (Coef:0.78, 95%CI: 0.33 to 1.22, *p* = 7 × 10^−7^ and Coef: 0.67, 95%CI: 0.07 to 1.27, *p* = 0.03, respectively).

*SLCO1B1**4, *SLCO1B1**19, and *SLCO1B1**20 were associated with decreased bilirubin and the remaining variants with increased bilirubin, and *SLCO1B1**5 had the highest risk (Coef: 1.8, 95%CI: 1.6 to 2.97, *p* = 6 × 10^−54^) (Appendix A).

### 2.2. General Practice (GP) Prescribing and Statin-Related Adverse Outcomes

**Muscle symptoms**: In total, 3.4% of participants with GP-prescribed statin data had GP-diagnosed muscle symptoms. Of these, 25% were *SLCO1B1**15 homozygotes and 9.9% were *SLCO1B1**20 homozygotes (Table 3). *SLCO1B1**15 carriers had increased risk of overall muscle symptoms versus non-carriers (HR: 1.16, 95%CI: 1.04 to 1.3, *p* = 0.01), of which carriers on statin for ≥5 years had a risk of 1.22 (95%CI: 1.07 to 1.39, *p* = 0.003). With 5+ years of statin prescription, *SLCO1B1**20 carriers had increased risk (HR: 1.25, 95%CI: 1.05 to 1.48, *p* = 0.01) vs. non-carriers (Figure 1, Appendix A). Using TWIST [27], we found that the RGMTE (robust genetically moderated treatment effect) and MR estimates were sufficiently similar to combine (Q-statistic *p* > 0.05) into a more precise estimate of GMTE: for *SLCO1B1**15, the per-year risk difference was 0.08% (*p* = 8 × 10^−3^). To illustrate this effect, we multiplied by the 80,685 *SLCO1B1**15 carrier patient years in the data; if we could reduce the *SLCO1B1**15 carrier’s risk by this amount, then the total cases would reduce by 65 (1457 to 1392), corresponding to a 4.5% reduction in muscle symptoms experienced by statin users, and 17.6% in the *15 carriers alone (369 to 304).

For *SLCO1B1**20, the per-year risk difference was 0.05% (*p* = 3.5 × 10^−2^). To illustrate this effect, we multiplied by the 34,076 *SLCO1B1**20 carrier patient years in the data; if we could reduce the *SLCO1B1**20 carrier’s risk by this amount, then the total cases would reduce by 17 (572 to 555), corresponding to a 3% reduction in muscle symptoms experienced by statin users, and 10.3% in the *20 carriers alone (165 to 148).

**Discontinuation**: Patients with *SLCO1B1**5 and *SLCO1B1**15 variants had slightly increased risk of discontinuation of statin therapy (HR: 1.12, 95%CI: 1.01 to 1.25, *p* = 0.04 and HR: 1.05, 95%CI: 1 to 1.1, *p* = 0.04, respectively) versus their non-carrier counterparts (Figure 2, Appendix A). *5 was found to increase risks for women (HR: 1.19, 95%CI: 1.03 to 1.37) but not for men (HR: 1.05, 95%CI: 0.92 to 1.20) [11]. No other variants increased risks in women in this study. While *SLCO1B1**5 was not significant in men, *SLCO1B1**15 increased risk in men (HR: 1.07, 95%CI: 1 to 1.13, *p* = 0.04) versus non-carriers (Appendix A).

**CHD and ischemic stroke**: During the statin prescription period, 22.8% of patients had myocardial infarction or angina, while 2.5% had ischemic stroke. Carriers of *SLCO1B1**37 had reduced CHD risk versus non-carriers (HR: 0.94, 95%CI: 0.89 to 0.99, *p* = 0.02) (rates: 21.8% versus 23.1%, respectively). Other variants showed negative trends but no significant associations with CHD or stroke. The lowest stroke rates were for *SLCO1B1**5 carriers (2.38%), and the highest rates were for *SLCO1B1**20 carriers (2.8%) (Appendix A).

### 2.3. Statin Gene Risk Score

We calculated the gene scores GRS-1 (“High risk” rs4149056 -*5 CC allele, or CT for *5 and rs2306283-AA (*20), rs11045819-CC (*4), or rs34671512-AA (*19), as reported by Bigossi et al. [19]) and GRS-2 (using rs2306283-GG instead of the AA allele) (see methods for details).

At baseline assessment, in participants self-reporting statin use with LDL measured (n = 61,898), the “high risk” *SLCO1B1* GRS-1 patients (n = 3667) did not have significantly higher LDL (Coef: 0.028 mmol/L: 95%Cis −0.001 to 0.061, *p* = 0.09) compared to the “low risk” GRS-1 group, whereas “high risk” *SLCO1B1* GRS-2 patients (n = 2564) had significantly raised LDL (Coef: 0.062 mmol/L: 95%Cis 0.022 to 0.101, *p* = 0.002) compared to the “low risk” GRS-2 group.

In the analysis of GP-prescribed statins, “high risk” GRS-2 patients had an increased likelihood of developing GP-diagnosed muscle symptoms (HR: 1.13, 95%CI: 1.02 to 1.27, *p* = 0.02), compared to “low risk” patients. This estimate was larger than the estimate in *5 CC homozygotes alone (HR: 1.09, 95%CI: 1.00 to 1.18, *p* = 0.048). Similarly, “high risk” GRS-2 patients had increased rates of discontinuation (HR: 1.07, 95%CI: 1.02 to 1.11, *p* = 0.005) compared to “low risk” patients, with estimates larger than those for *5 C allele carriers (HR: 1.04, 95%CI: 1 to 1.08, *p* = 0.02), but smaller than that for *5 CC homozygotes (HR: 1.12, 95%CI: 1.01 to 1.26, *p* = 0.04), compared to non-carriers.

### 2.4. Rare Variants in SLCO1B1 Identified by Exome Sequencing

We studied 47 rare variants predicted by the VEP (https://www.ensembl.org/info/docs/tools/vep/index.html, accessed on 1 February 2023) tools to be missense or to have other moderate/high-impact consequences for OATP1B1 function, and that were present in at least three participants taking statins at the baseline assessment (n = 61,893). Two of these were associated with increased LDL at baseline in linear regression models adjusted for age, sex, and five principal components (FDR-adjusted *p*-value < 0.05). These rare variants had substantially larger effect sizes in comparison to *SLCO1B1**5, where the per-allele increase in LDL was 0.08 mmol/L, *p* = 6 × 10^−6^: rs374113543 (MAC = 4) increased LDL by 1.05 mmol/L (*p* = 0.001) and rs373327528 (MAC = 9) by 0.7 mmol/L (*p* = 0.003). Two further variants were nominally associated with LDL (FDR *p* > 0.05): rs960742177 (MAC = 7. Coef: 0.7, *p* = 0.009) and rs780911188 (MAC = 4. Coef: 0.65, *p* = 0.047). None of the rare variants were associated with LDL in untreated people in the TWIST analysis [27] (see Figure 3 and Appendix A for details). The variants were not in linkage disequilibrium (LD) (R^2^ < 0.01) and were independent in the multivariable analysis.

In the combined analysis, 24 participants carried at least one allele of the four LDL-increasing rare variants (45.8% female, mean age: 61.64 [SD 3.5]). Those 24 participants had higher LDL (0.76 mmol/L, 95%CI: 0.48 to 1.03, *p* = 7 × 10^−8^) compared to non-carriers; 45.8% had clinically high LDL (>3.5 mmol/L) compared to 11.2% in non-carriers (Appendix A for the descriptive table of rare variant and SLCO1B1*5 carriers). In the stratified analysis, the effect was substantially greater in men vs. women (Coef: 1.77, 95%CI: 1.21 to 2.33, *p* = 5.6 × 10^−10^; Coef: 0.39, 95%CI: −0.23 to 1, *p* = 0.2). A total of 62.5% of patients carrying the variants reported experiencing pain at the baseline assessment, with the majority being men (66.6%). Male participants carrying the LDL-increasing rare variants had increased risk for experiencing pain while on statins compared to non-carriers (odds ratio (OR) = 1.39, 95%CI: = 1.06 to 1.82, *p* = 0.002). It was not significant for women (OR = 1.02, 95%CI: 0.8 to 1.36, *p* = 0.91).

See Appendix A for details.

See Appendix A for the other biomarker associations in rare variants.

### 2.5. SLCO1B1 “High Risk” Gene Score plus Rare Variants

In the combined analysis, the number of *SLCO1B1* “high risk” patients increased from 2564 (GRS-2 in the baseline LDL analysis) to 2588 when including those carrying LDL-increasing rare variants. The overall association with LDL increases modestly in this group (compared to GRS-2 associations), due to the small difference in numbers (e.g., GRS-3 LDL coefficient: 0.055, *p* = 3 × 10^−5^; GRS-2 LDL coefficient: 0.049, *p* = 2 × 10^−4^). The associations with other statin outcomes were not notably different when incorporating rare variants into the common variant *SLCO1B1* high-risk gene score.

The numbers of patients prescribed other statins were too limited to analyze. Details on the numbers can be found in [11].

## 3. Discussion

Common *SLCO1B1* variants are reported to affect statin response, but the evidence on clinical outcomes is conflicting. We used whole-exome sequencing data and studied common and rare *SLCO1B1* genetic variation in a large cohort of UK Biobank statin patients to clarify risks in a community cohort. The *SLCO1B1**5 variant alone and *SLCO1B1**15 haplotype (*5 and *37 inherited together) increased LDL at UK Biobank baseline assessment and increased rates of statin prescription discontinuation in the GP data. *SLCO1B1**15 and *20 also increased the risk of GP-diagnosed muscle symptoms during statin treatment; we estimated that genotype-guided prescribing may prevent 17.6% of diagnosed muscle symptoms during statin treatment in *15 carriers, and 10.3% in the *20 carriers. This illustrates the absolute effect of these pharmacogenetic variants on SAMS events. The evidence for other common variants was weaker and less reliable. We also identified two rare mutations with large effects on LDL in statin users: 45.8% of the rare allele carriers had clinically high LDL (>3.5 mmol/L) during statin treatment, compared to 11% of non-carriers. Combining rare variants with common-variant gene risk scores to identify patients at “high risk” of statin-associated muscle symptoms (SAMS) could further improve LDL control, though replication is needed, as is cost–benefit analysis, given the predominantly modest effect sizes observed.

rs11045819 (Pro155Thr) and rs34671512 (Leu643Phe) increase simvastatin bioavailability [18], suggesting GoF effects. In our study these variants improved the estimates for statin outcomes in UK Biobank patients when combined into a gene risk score (GRS); in “high risk” patients (i.e., carrying a *5 risk allele and no GoF alleles), GP-diagnosed muscle symptoms were higher (HR: 1.13, *p* = 0.02) than the estimate for *5 CC allele carriers alone (HR: 1.09, *p* = 0.048), compared to non-carriers. These findings partially replicate the Bigossi et al. [19] GRS in 15,378 statin users, where incorporating GoF variants into the GRS improved the prediction of SAMS. The authors also included *SLCO1B1**37 in the GRS, defining the G-allele as protective. Other studies report that the G-allele protects against statin-related adverse events in 4196 [6] and 15,378 statin patients [19]. In our analysis, we found that the *37 G-allele *increased* the risk of statin adverse outcomes when included in the GRS. This difference between our study and Bigossi et al. warrants further investigation; PharmGKB reports that *37 is a “normal” function variant, rather than protective [3], due to studies suggesting the *37 effect varies with the presence of *5 [28]. We found that neither *5 nor *37 alone is associated with GP-diagnosed muscle symptoms, yet *15 carriers (the combination) had higher risk, suggesting—at least in UK Biobank participants—that the *37 G-allele in combination with *5a is the risk-increasing allele. More work is required to understand the context-dependent effects of *37. Additionally, we found that *37 homozygotes have a nominally decreased risk of CHD during dCCB treatment (HR 0.94, *p* = 0.02), compared to participants with no *37 alleles. This may be an anomalous result, given that we found no evidence that *37 homozygotes correlate with cholesterol, GP-diagnosed muscle symptoms, or treatment discontinuation.

For *SLCO1B1**15 (*5 and *37 in the same individual), PharmGKB reports high-level evidence for pharmacokinetic relationships, acting similarly to *5, noting limited evidence of an effect on clinical outcomes [29,30]. Our findings extend this, showing that although estimates for the effect of *5 on LDL are greater than those for *15, the *15 individuals (a subset of *5 who also carry *37-G) had higher rates of GP-prescribed statin discontinuation and GP-diagnosed muscle symptoms. However, given the weight of prior evidence, the effect sizes and significance we observed are modest. We found that *5 and *15 are associated with lower vitamin D levels. This is consistent with a previous meta-analysis, where low vitamin D was associated with SAMS [24], and supports a causal role of statin pharmacogenetics in lowering vitamin D. *SLCO1B1**20 was associated with increased vitamin D (though the association was only borderline statistically significant: *p* = 0.03) and SAMS, contrary to the other genotypes. This could be due to the complexity of SAMS’s mechanisms; we hypothesize that the reduction of unintended targets for HMG CoA Reductase inhibition, such as dolichols, ubiquinone, and prenylated proteins [31], may be the reason for the inverted *SLCO1B1**20 effect, as it has been found to increase this activity, albeit with limited evidence [3]. Further investigations are required.

Two studies of 717 and 699 patients report large effects of rare variants on relevant phenotypes [19,28]. Consistent with this, we identified two rare variants that substantially increased LDL levels compared to common variants. This is a trend often seen in human genetics, due to selection pressure against large-effect variants that perturb metabolism. These variants were not associated with LDL in untreated people (the effect on LDL is therefore likely pharmacogenetic on statins). Including them in the GRS increased the overall effect size, and suggests that identifying specifically high-risk individuals for monitoring and intervention might be possible. In a recent study, both rare variants (rs374113543-rs373327528) were reported to not significantly affect OATP1B1 protein expression [21], though transporter activity for these variants was not assessed. These variants have not otherwise been previously reported and should be interpreted with caution until replicated.

We also investigated other biomarkers at the UK Biobank baseline. Bilirubin hepatic transport is carried out by organic anion-transporting polypeptides (OATP) including OATP1B1 (coded by *SLCO1B1*), and its levels are increased by atorvastatin [26]. In our study, *SLCO1B1* genetic variants affected bilirubin levels in untreated UK Biobank participants and those treated with statins. Higher direct bilirubin was associated with ischemic stroke (in 275 stroke patients) [25], but we did not find associations between *SLCO1B1* variants and ischemic stroke, possibly due to few UK Biobank patients receiving diagnoses of stroke.

Our study has several strengths. To our knowledge, it is the largest analysis of exome sequencing data in statin patients in a general-population cohort with linked medical-record follow-up over decades. We provide clinical evidence on statin outcomes affected by *SLCO1B1* common variants, haplotypes, and rare variants. We also extend a recent [19] *SLCO1B1* ‘gene risk score’ and confirm that combining *SLCO1B1**5 and GoF can improve estimates of adverse outcomes. We used the TWIST framework to estimate the genetically moderated treatment effect (number of diagnoses attributed to the genotype) at the group level. This has multiple strengths, such as including information from untreated people, and combining estimates from independent statistical methods. Whilst this framework allows us to conclude in this study that a percentage of adverse events in patients on atorvastatin and simvastatin could have been avoided if patients with the genotype received the normal effect of the treatment, it cannot yet be used to make treatment recommendations for individual patients. This would require an in-depth clinical assessment of the various treatment options available to each individual and is a question under active consideration as part of ongoing research.

This study has several limitations. UK Biobank is a volunteer cohort, and participants were healthier than the general population at baseline [32], and recall bias may affect our findings. Using electronic medical records after baseline assessment helps alleviate this limitation. The ascertainment of statin-related myopathy and related diagnoses was limited due to the lack of available diagnostic codes in the GP record. SAMS’s prevalence may be overestimated in cohort studies (17%), with evidence from RCTs suggesting much lower true prevalence (4.9%) after the blinding of participants [33]; in our ICD10 code-based data, we identified 3.4% prevalence, which is closer to this estimate from RCTs. Furthermore, genotype did not increase the risk of SAMs in untreated people, consistent with the statin-specific pharmacogenetic mechanism. The GP-ascertained outcomes had limited power to detect rare-variant associations, compared to quantitative baseline measures. Inconsistency in rare-variant associations necessitate further studies to establish a comprehensive understanding. We were not able to examine statin doses and other statins due to the limited availability of reliable data.

Our results show that, although patients carrying *SLCO1B1* variants are more likely to be diagnosed with muscle pain and discontinue statin prescriptions, they do not have increased cardiovascular disease risk. This may be due to GPs overcoming the effect of *SLCO1B1* variants and achieving target LDL levels; however, we observed large effects on LDL at the UK Biobank baseline; therefore, more data are needed. UK Biobank includes generally healthy individuals from socioeconomically advantaged backgrounds [32]. In the general population, LDL monitoring may differ if patients do not see their GPs to report symptoms, and thus, adverse events in higher-risk groups might be more detrimental. Our study is therefore potentially significant in the context of real-world healthcare practices and may have population-wide implications. We hope that our study contributes to informing how genotype information is used in routine clinical practice, once genotype information is known for many patients; for example, following the implementation of Our Future Health [34], where 5 million UK adults will have genotype information and consent for actionable results to be referred to their GP, the rate of avoided discontinuation could rise dramatically. Our findings of modest effects of SLCO1B1 variants on the outcomes studied do not support specific genotyping.

In conclusion, common *SLCO1B1* variants are associated with adverse outcomes in 64,000 statin patients from a largescale general-population cohort. Rare variants with large effects are also identified, as are improved estimates for the *SLCO1B1* gene risk score, though replication is needed.

## 4. Materials and Methods

### 4.1. UK Biobank (UKB)

UK Biobank comprises 503,325 community-based volunteers aged 40–70 years enrolled in Wales, Scotland, and England in 2006–2010. The Northwest Multi-Center Research Ethics Committee approved the collection and use of UKB data (Research Ethics Committee reference 11/NW/0382). Participants gave informed consent for the use of their data, health records, and biological materials for health-related research purposes. Access to UKB was granted under Application Number 14631. Baseline assessment included the collection of demographic, lifestyle, and health information via questionnaires, and anthropometric measures. Participants provided blood samples for genetic and biochemical analyses. This study comprises two distinct analyses: (1) data from baseline assessment, and (2) using the linked GP (General Practice) data available in 230,096 (45.7%) participants (Figure 3).

GP data are up to August 2016 (TPP system supplier in England) and September 2017 (EMIS/Vision system in Wales). Drug name, quantity, date of prescription, and drug code (in Clinical Read version 2, British National Formulary [BNF] or DM + D [Dictionary of Medicines and Devices] format, depending on the provider) are available. Prescribing records included 10, 20, 40, and 80 mg simvastatin (zocor, simvastatin, simvadol) and 10, 20, 40, 60, and 80 mg atorvastatin (atorvastatin, lipitor) prescriptions. All available formulations were considered regardless of dose due to the limited availability of reliable dose data. The censor date is the date of withdrawal (deletion from the GP list, where available) or 28 February 2016 if the date of withdrawal is not available (i.e., still registered in an available clinic). Data after 28 February 2016 were incomplete (see UK Biobank document [35]; details are described elsewhere [11]).

### 4.2. Whole-Exome Sequencing (WES)

Detailed methods for WES analysis in UK Biobank were published by the Regeneron Genetics Center [36]. In brief, participant DNA was enzymatically shared with ~200 base pairs to create a DNA library, with a unique 10-base-pair barcode added to facilitate multiplexed exome capture with a modified IDT xGen probe library. DNA was PCR-amplified and sequenced using 75-base-pair paired-end reads on an Illumina NovaSeq 6000 platform (Illumina, Inc., San Diego, CA, USA). Variant calling and quality control (QC) were performed centrally (described previously [37]). ‘BWA MEM’ mapped reads to the hg38 reference genomes, and ‘WeCall’ identified small SNP and indel variants. Only called variants that have passed rigorous QC are available to researchers. Data are available on 469,914 participants.

Due to known differences in allele frequencies and pharmacogenetic effects across diverse ancestral groups, we focused on the participants genetically similar to the 1000 Genomes European (EUR) reference population (95% of UK Biobank participants), identified by principal component analysis (described previously [38]).

### 4.3. SLCO1B1 Variants

We identified previously reported variants and haplotypes with the “star” nomenclature [39], such as *SLCO1B1**5 and *SLCO1B1**15 (both *5 and *37—Figure 4 for details). In statin patients at baseline, the following *SLCO1B1* variants were observed: *15 (frequency = 23.7%), *14 (24%), *37 (15.1%), *20 (10.1%), *4 (2.4%), *5 (2.2%), *19 (0.26%), *46 (0.11%), and *45 (0.11%—heterozygotes). *31 and *47 were not observed due to the small numbers of carriers (Figure 4).

To identify novel variants with a potentially high impact on OATP1B1 protein function, we used the Variant Effect Predictor (VEP) [40] to estimate their effect on *SLCO1B1* transcripts, protein sequence, and regulatory regions (see Appendix A for VEP output). We investigated all *SLCO1B1* variants in the UKB WES data between GRChg38 coordinates chr12:21131194 and chr12:21239796 (https://www.ncbi.nlm.nih.gov/gene/10599, accessed on 1 February 2023). There were 1520 unique variants, of which 489 were identified as missense variants, and 618 were predicted to have a moderate or high impact: 185 unique variants were found in statin patients at baseline and 199 in patients for whom statin were prescribed by their GP, of which 51 had a minor allele count over 3, of which 4 were common variants (MAF > 0.1%) and 47 were rare-variants (Figure 4 for the flowchart).

We calculated the MAC (minor allele count: total number of minor alleles carried by patients) and MAF (minor allele frequency: the proportion of all alleles in the participants that were the minor allele) for each variant. We included variants in the single-variant analyses if their MAC > 3 (Appendix A for MAC and MAF for included variants).

### 4.4. Statin Gene Risk Score (GRS)

We created two GRSs.

(1) “High risk” patients are homozygous for the *5 C allele, or are heterozygous for *5 and do not carry protective alleles for *37 (i.e., are rs2306283-A homozygotes), Pro155Thr (i.e., are rs11045819-C homozygotes), or Leu643Phe (i.e., are rs34671512-A homozygotes), as reported previously [19]. All remaining allele combinations are defined as “low risk”.

(2) We found that for *37, it is the G-allele that has similar trends to the detrimental *5 allele in analyses of LDL, GP-diagnosed muscle symptoms, treatment discontinuation, and hospital-diagnosed coronary heart disease (CHD) (in contrast to the previous study [19]).

We therefore analyzed the GRS in two ways: (1) as reported by Bigossi [19] and (2) using rs2306283-G homozygotes in the “high risk” group (Table 4).

### 4.5. Outcomes

We examined outcomes in two categories: baseline and GP-linked outcomes. Where data were available, we included traits linked to *SLCO1B1* in the GWAS Catalog [41].

(1)Baseline

LDL, triglycerides, total cholesterol, direct bilirubin, C-reactive protein (CRP), alanine aminotransferase (ALT), cystatin C, vitamin D, and Glycated hemoglobin (Hba1c) were included (mostly from GWAS catalogue for SLCO1B1). We also extracted ‘pain experienced in last month’ variable (Data-Field 6159: head or shoulder pain, back pain, hip pain, knee pain, or pain all over body).

(2)General Practice (GP) data

We utilized ICD-10 codes for conditions related to myopathy, myositis, myalgia, or rhabdomyolysis. Discontinuation was defined when patients had their last prescription at least 3 months prior to the censoring date (removal from GP list) or by 28 February 2016, if no deduction date was available. We utilized ICD-10 codes for conditions related to myopathy, myositis, myalgia, or rhabdomyolysis (G72.0, G72.8, G72.9, M60.8, M60.9, M79.1, M62.82) and converted them to the Read codes used in UK primary care records, utilizing diagnostic code maps provided by UKB.

We captured cardiovascular events from hospital admissions data, covering up to 15 years of follow-up (HES data in England until September 2021, with data from Scotland censored in August 2020 and Wales in February 2018). We identified diagnoses of CHD, including myocardial infarction and angina, using ICD-10 codes I20*, I21*, I22*, I23*, I24*, and I25*. Additionally, diagnoses of ischemic stroke were identified using ICD-10 code I63*.

### 4.6. Statistical Analysis

(1)Baseline assessment data for all variants

We conducted linear regression analyses to estimate associations between biomarkers and *SLCO1B1* variants in statin users aged 40+, adjusting for sex, age, assessment center, and principal components of genetic ancestry 1–5 to account for population substructure. We also tested associations between self-reported pain and LDL-increasing rare variants.

(2)GP data for common variants

We performed time-to-event analyses using electronic medical record data (primary- and secondary-care-linked) for the common (*) variants. We included patients who were 40+ years old at the time of their first statin prescription and had a minimum of two months’ prescriptions during the treatment period. Cox’s proportional hazards regression models were used to assess the associations of GP-diagnosed muscle symptoms, discontinuation of statin therapy, hospital inpatient-diagnosed CHD, and ischemic stroke with the * variants, adjusting for sex, age at first statin prescription, and principal components of genetic ancestry (Appendix A for details).

(3)TWIST

For the significant LDL-altering variants, we used our novel pharmacogenetic causal inference approach “TWIST” (Triangulation with a Study [27]) to estimate the population-average effect on outcomes if all *SLCO1B1* variant carriers could experience the same treatment effect as non-carriers. This extends the well-established idea of estimating avoidable events in pharmacogenetics by incorporating information from untreated individuals and triangulating evidence for multiple statistical methods. Briefly, the method uses regression models to test several assumptions common to pharmacogenetic analysis, primarily that the genetic variants do not predict whether an individual receives statin treatment; are not associated with any measured confounders predicting statin use or the studied outcome; and only affect the outcome through the interaction with statins (see [27] for details). The method robustly estimates the ‘Genetically Moderated Treatment Effect’ (GMTE): see Appendix A for details. We used the {twistR} R package v0.1.4 (https://github.com/lukepilling/twistR, accessed on 1 March 2023).

## Figures and Tables

**Figure 1 ijms-25-04426-f001:**
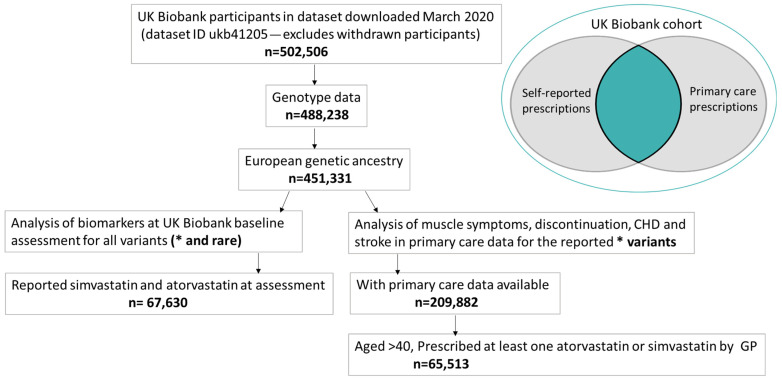
Cohort flowchart. CHD: coronary heart disease; GP: general practitioner. We conducted baseline analyses for common variants, including haplotypes, and rare variants. Due to the small number of events in rare-variant-carrier patients, we conducted primary care analyses in patients carrying common variants. * refers the variants described with star (e.g., *5).

**Figure 2 ijms-25-04426-f002:**
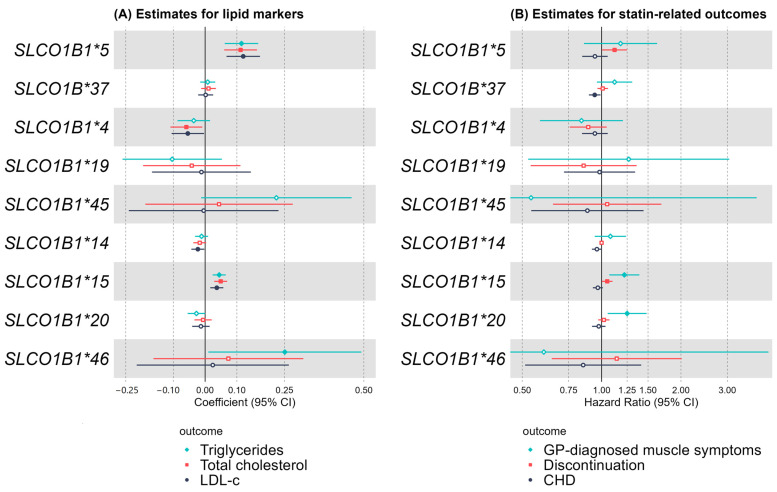
Associations between common variants, including haplotypes and statin-related outcomes. (**A**) Associations in UK Biobank baseline assessment data. Coefficient = difference in LDL (mmol/L), total cholesterol (mmol/L), and triglycerides (mmol/L) between analysis group and reference group. First four variants are common ‘*’ variants, *45 is a known ‘*’ variant, and the remaining variants are haplotypes. Models were adjusted for sex, age at baseline assessment, and genetic principal components. (**B**) Associations in UK Biobank primary care data. Muscle symptoms refer GP-diagnosed muscle symptoms, including myopathy, myalgia, myositis, or rhabdomyolysis. The table shows the results for patients who had been on statin prescriptions for at least 5 years. Discontinuation means patients had their last prescription at least 3 months prior to the censoring date (removal from GP list) or by 28 February 2016, if no deduction date was available. CHD: hospital inpatient coronary heart disease diagnosis. Models were adjusted for sex, age at first prescription, and genetic principal components.

**Figure 3 ijms-25-04426-f003:**
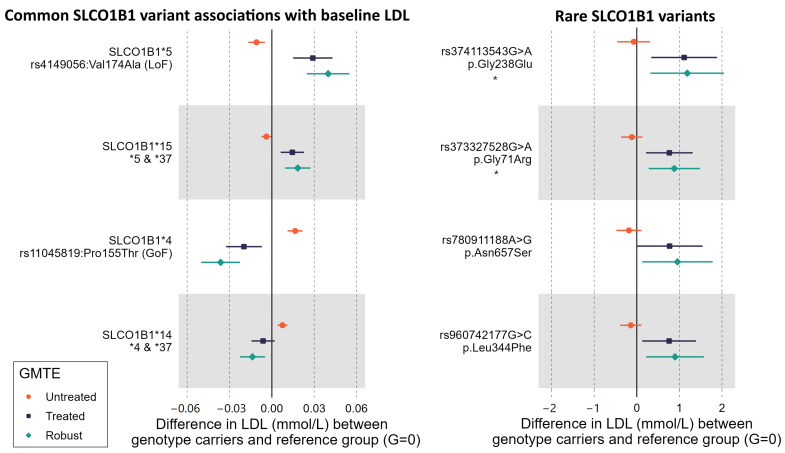
TWIST results demonstrating the pharmacogenetic nature of the studied variants. GMTE = genetically moderated treatment effect (Untreated = association between the variant and LDL in individuals not treated with statins at the time of assessment; Treated = association between the variant and LDL in individuals who self-reported statin treatment at the time of assessment; Robust = the ‘treated’ association accounting for any effect in ‘untreated’ individuals). Rare variants: Gly238Glu and Gly71Arg were significantly associated with LDL in treated individuals after multiple testing adjustment (FDR < 0.05, indicated with a ‘star’). The other rare variants shown were only nominally associated. See Appendix A for details.

**Figure 4 ijms-25-04426-f004:**
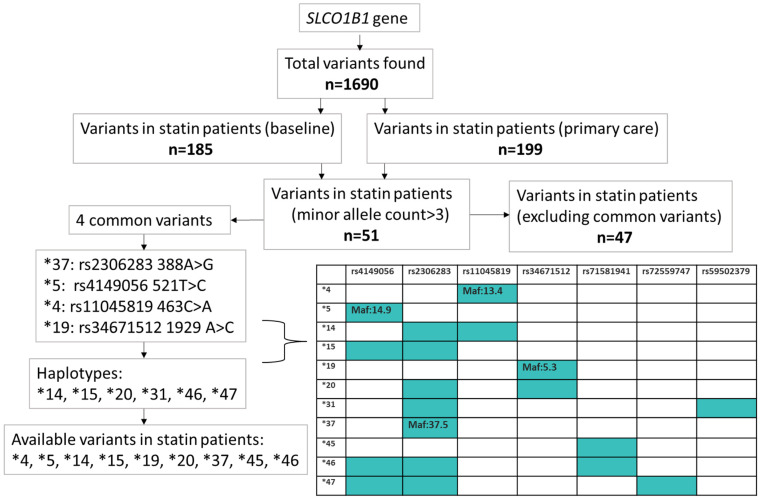
*SLCO1B1* variants flowchart. We identified 1690 SLCO1B1 variants in whole-exome sequencing data through a variant effect predictor (VEP); of those, we filtered for high–moderate impact consequences in patients who reported statin use at baseline (n = 185) and had a GP statin prescription in their primary care records (n = 199). We included variants with mac > 3 (n = 51); of those 51, 4 were common variants. We present the minor allele frequencies of 4 common variants in the table below. We also present 6 haplotypes (represented by a ‘star’) of the 4 common variants in the table.

**Table 1 ijms-25-04426-t001:** Characteristics of UK Biobank participants.

		Baseline Data
**Sex**	N Female	26,185 (38.7)
**Age**	Mean (SD)	61.5 (6)
	min-max	40–70
	Median	63
**Simvastatin**	N (%)	51,147 (75.5)
**Weight**	Mean (SD)	84.3 (16.5)
**BMI**	Mean (SD)	29.4 (5)
**LDL**	Mean (SD)	2.75 (0.67)
**LDL, n > 2.86**	N (%)	28,450 (42)
	N female (%)	12,622(44.3)
**Total cholesterol**	Mean (SD)	4.6 (0.9)
**Triglycerides**	Mean (SD)	1.9 (1.1)
**Direct Bilirubin**	Mean (SD)	9.5 (4.5)
**ALT**	Mean (SD)	27.1 (14.5)
**CRP**	Mean (SD)	2.8 (4.7)
**Cystatin c**	Mean (SD)	0.9 (0.2)
**Vitamin D**	Mean (SD)	49.7 (21.6)
**HbA1c**	Mean (SD)	40.5 (10.2)
	**Primary care data**	
**Sex**	N Female	27,299 (42)
**Duration on statins^ (year)**	Mean (SD)	6.7 (4.7)
	min-max	1 to 25
**Number of prescriptions per year**	Mean (SD)	9 (5)
**Muscle diagnoses ^1^ prior to statin ^**	N (%)	1308 (2)
**MI/angina diagnoses ^2^ prior to statin ^**	N (%)	6718 (10.4)
**Stroke diagnoses ^2^ prior to statin ^**	N (%)	855 (1.3)
**Muscle diagnoses ^1^ after first statin ^**	N (%)	2242 (3.5)
**MI/angina diagnoses ^2^ after first statin ^**	N (%)	14,751 (22.8)
**Stroke diagnoses ^2^ after first statin ^**	N (%)	1629 (2.5)
**Discontinuation ever**	N (%)	13,889 (22.4)

Abbreviations: BMI, body mass index; LDL, low-density lipoprotein cholesterol; ALT, Alanine aminotransferase; CRP, C-reactive protein; MI, myocardial infarction. ^ Simvastatin or atorvastatin prescription. ^1^ Primary care-diagnosed muscle symptoms (myopathy, myositis or myalgia). ^2^ Hospital inpatient diagnosis.

**Table 2 ijms-25-04426-t002:** Baseline LDL levels associations with star and common variants.

Allele						
*Common variants*	N	%	Coef	95%	CI	*p*
***4-CC**	45,798	75.58	*ref*			
***4-CA**	13,324	21.99	−0.02	−0.03	−0.009	8 × 10^−4^
***4-AA**	1475	2.43	−0.03	−0.067	−0.0008	0.045
***5-TT**	44,821	72.47	*ref*			
***5-TC**	15,640	25.29	0.02	0.01	0.03	4 × 10^−4^
***5-CC**	1383	2.24	0.08	0.05	0.12	6 × 10^−6^
***37-AA**	24,413	40.03	*ref*			
***37-AG**	27,340	44.83	0.01	−0.01	0.02	0.32
***37-GG**	9235	15.14	2 × 10^−3^	−0.01	0.02	0.77
***19-AA**	55,469	89.69	*ref*			
***19-AC**	6223	10.06	−0.01	−0.03	0.01	0.18
***19-CC**	154	0.25	−0.01	−0.12	0.09	0.83
** *Haplotypes ^* **	**N**	**%**	**Coef**	**95%**	**CI**	** *p* **
**No *14**	23,870	39.91	*ref*			
** *14**	14,181	23.71	−0.01	−0.03	0.00	0.05
**No *15**	22,015	36.13	*ref*			
** *15**	14,454	23.72	0.03	0.01	0.04	3 × 10^-4^
**No *20**	24,254	39.80	*ref*			
** *20**	6173	10.13	−0.01	−0.03	0.01	0.42
**No *45**	60,852	99.89	*ref*			
** *45 °**	67	0.11	−0.01	−0.17	0.15	0.90
**No *46**	21,707	36.2	*ref*			
** *46**	65	0.1	0.002	−0.01	0.01	0.70

Coef: = difference in LDL (mmol/L) between analysis group and reference group; ref = Reference group for analysis. Linear regression analysis of LDL at UK Biobank baseline assessment, in up to 64,000 participants who self-reported taking simvastatin or atorvastatin and are genetically similar to the 1000 Genomes European reference population, adjusted for age, sex, and genetic principal components. ^ for haplotypes, the reference group are participants without any copies of the constituent variants (e.g., for *15 controls carry no copies of *5 or *37). % = percentage of 64,000 statin-users who are the indicated genotype/haplotype. ° = *45 is not a haplotype, ° refers the heterozygotes.

**Table 3 ijms-25-04426-t003:** Percentage of *SLCO1B1* genotype carriers with and without SAMS.

		*4		*5		*14		*15		*20		*37		*45		*19	
		%	n	%	n	%	n	%	n	%	n	%	n	%	n	%	n
**SAMS**	wild	72.36	1542	67.81	1445	35.48	756	33.36	711	36.23	772	36.60	780	94.79	2020	85.83	1829
	heterozygotes	20.13	429	26.09	556	36.65	781	36.60	780	48.71	1038	43.55	928	0.05	1	9.95	212
	homozygotes	1.88	40	2.39	51	21.16	451	25.06	534	9.90	211	14.88	317	0.00		0.33	7
**No SAMS**	wild	71.98	44,126	69.70	42,730	37.84	23,194	34.78	21,319	38.43	23,559	38.67	23,707	94.82	58,129	86.57	53,070
	heterozygotes	20.17	12,364	24.34	14,920	33.81	20,728	37.63	23,069	46.99	28,808	42.02	25,762	0.11	70	9.48	5810
	homozygotes	2.24	1373	2.26	1388	21.48	13,169	22.45	13,763	9.45	5792	14.26	8741	94.82		0.28	169

SAMS excluding existing muscle symptoms. N = 2131 with SAMS and 61,303 without.

**Table 4 ijms-25-04426-t004:** Statin Gene Risk Score (GRS).

		*5	*37	rs11045819	rs34671512
	High risk	CC			
**GRS1**	High risk	TC	AA	CC	AA
	Low risk	all remaining			
	High risk	CC			
**GRS2**	High risk	TC	GG	CC	AA
	Low risk	all remaining			

*37 G-allele has similar trends to the detrimental *5 allele in analyses of LDL, GP-diagnosed muscle symptoms, treatment discontinuation, and hospital-diagnosed CHD. We therefore analyzed the GRS in two ways: GRS1 as reported by Bigossi [19] and GRS2 using rs2306283-G homozygotes in the “high risk” group.

## Data Availability

Participant-level data are available on application to the UK Biobank (www.ukbiobank.ac.uk/register-apply, accessed on 20 March 2024). The TWIST R package is available on GitHub (https://github.com/lukepilling/twistR, accessed on 20 March 2024) but contains no individual-level data.

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
