# Peer review of "SLCO1B1 Exome Sequencing and Statin Treatment Response in 64,000 UK Biobank Patients"

_ijms, 2024, doi:10.3390/ijms25084426_

Round 1

Reviewer 1 Report

Comments and Suggestions for Authors Turkmen et al attempted to analyze 51 variants in the solute carrier organic anion transporter family member 1B1 as relevant to statin treatment, circulating LDL levels, and adverse outcomes associated with muscle symptoms. Two major variants reported previously were confirmed, several minor and rare variants were added to the list of potentially informative outcomes. The study is based on a large data sample of UK Biobank exome sequences and follow up medical records.      Major concerns:   Add a sentence to the abstract that explains what SLCO1B1 is, its function including drug transport, and what is the importance of studying this gene   References not in order, and not in correct format    Move descriptions listed under the figures into figure legends   Provide Figure 3 in adequate quality, fonts are too small to read    Line 401, “relatively young and healthy” is an incorrect statement given the median 63 year old subjects   References/citation list is very poor with many citations lacking correct information    Minor concerns:   Line 13, fix fonts   Line 15, include full names of the gene, solute carrier organic anion transporter family member 1B1   Line 17, what "in up to 64,000 patients” means?    Line 19, include full name of the protein, membrane organic anion OATP1B1   Line 23, explain what GP-diagnosed is   Edit the funding and contribution sections for clarity 

Reviewer 2 Report

Comments and Suggestions for Authors

This is a kind of the follow up of the paper by the same authors dedicated to TWIST package. Consequently it can be properly understood in this context after readers check the corresponding R package and the previous publication. Nevertheless, this paper is useful in all respects and is suitable for the publication after some minor adjustments.  

1. There are several names of prescribed statins, authors mention two of them in selected patients group. I wish authors could add some  info on other frequently prescribed statins (like for instance rosuvastatin ) and comment briefly whether there is a difference in pharmacological effects and adverse symptoms or not. In any case please tell why those two and is there any difference?

2. When reporting biobank frequencies for SLCO1B1 gene, does it include all variants, or only allelic variants? Do authors include or ignore somatic variants (if they detected). Also related, when reporting allelic frequencies. How many alleles are included in the whole list as 100%?

3. Abbreviations should be explained at their first appearance, which is not the case. Please explain it properly and try to avoid excessive use of it.

4. Authors use hazard ration as a measure of the allelic effects. Can it be converted into the effect size? How big is the effect of the alleles *5 and *15 in the model predicting SAMS? How much of the  muscle syndrome (SAMS) is finally explained by their model, and what was included in the model? 

5. Data availability. Please comment which data are directly available in TWIST, which includes some kind of the statins treatment source as an example for the package.
